# First-Time Acute Lateral Patellar Dislocation in Children and Adolescents: What about Unaffected Knee Patellofemoral Joint Anatomic Abnormalities?

**DOI:** 10.3390/medicina57030206

**Published:** 2021-02-26

**Authors:** Rasa Simonaitytė, Saulius Rutkauskas, Emilis Čekanauskas, Liutauras Labanauskas, Vidmantas Barauskas

**Affiliations:** 1Department of Pediatric Surgery, Medical Academy, Lithuanian University of Health Sciences, LT-50161 Kaunas, Lithuania; emilis.cekanauskas@kaunoklinikos.lt (E.Č.); vidmantas.barauskas@kaunoklinikos.lt (V.B.); 2Department of Radiology, Medical Academy, Lithuanian University of Health Sciences, LT-50161 Kaunas, Lithuania; saulius.rutkauskas@gmail.com; 3Department of Children Diseases, Medical Academy, Lithuanian University of Health Sciences, LT-50161 Kaunas, Lithuania; liutauras.labanauskas@kaunoklinikos.lt

**Keywords:** acute patellar dislocation, patellofemoral joint, patella alta, femoral sulcus, trochlear dysplasia

## Abstract

*Background and Objectives:* Acute lateral patellar dislocation (LPD) is the most common acute knee disorder in children and adolescents, and may lead to functional disability. The purpose of this study was to identify key differences and correlations of the patellofemoral joint (PFJ) morphology between intact and contralateral injured knees in a first-time traumatic LPD population aged under 18 years. *Materials and Methods:* The data were gathered prospectively from a cohort of 58 patients (35 girls and 23 boys). The prevalence and combined prevalence of patella alta (PA) and trochlear dysplasia (TD) in both knees of patients were evaluated using X-ray by two radiologists. *Results:* The PFJ of patients’ intact knees had a lower rate of TD (1.72% vs. 5.2%) and a less common combination of PA with shallow femoral sulcus (SFS) (22.4% vs. 44.8%) but more frequent PA (62.1% vs. 41.4%) compared with their injured knees. We noted statistically significant positive correlations (SSPCs) between the femoral sulcus angle (FSA) and PA in patients with intact (*r* = 0.37; *p* < 0.005) and contralateral injured knees (*r* = 0.33; *p* < 0.05). *Conclusion:* There were SSPCs between the FSA and PA in both gender and age groups of patients with intact and contralateral injured knees. The SSPCs between the FSA and PA of intact knees were higher in the patients with a more dysplastic PFJ anatomy (PA and TD) of the injured knees as compared to patients with only PA of the injured knees.

## 1. Introduction

The first acute lateral patellar dislocation (LPD) is a traumatic and memorable event that typically occurs in active children or adolescents, and may lead to functional disability [1,2,3,4].

Normal stability of the patellofemoral joint (PFJ) is maintained by a complex interaction between soft tissues and the harmonious relationship of the patella with the femoral trochlear groove [5]. However, if extensive external forces twist, bend, or rotate the knee with a normal PFJ, an LPD can occur [6]. Acute LPD occurs more frequently in knees showing alterations of the normal knee anatomy that influence the function of the [1,2,3,4,5,6,7,8,9]. Based on the previous literature, more than half of all patients with patellar instability have one abnormal PFJ morphological factor, and about a third of them have two or more alterations in the surface geometry of the PFJ [5,10]. Patella alta (PA) and trochlear dysplasia (TD) are considered to be the primary anatomical conditions leading to lateral patellar maltracking and causing episodes of instability [2,9,10,11]. Previous studies have shown that persons with greater PFJ malalignment and instability have concomitant abnormal vertical position of the patella and flattened trochleae [2,9,10,11,12]. It is possible that there may be an integration of the relationship between the morphological development of the patella and the trochlea in the genome during the course of evolution, but it may also be a result of orthostatism and biped walking [13,14]. These provisions can explain demographic risk factors for the development of LPD such as gender, bilaterality, and a positive family history [8,9,15].

In addition, despite a large body of literature describing LPD risk factors and the PFJ aberrations causing its instability, the geometric morphological alterations of the contralateral intact knee PFJ are surprisingly ill-defined, especially in children and in the adolescent population suffering from acute first-time one-sided LPD.

To our knowledge, this is the first study to report the prevalence of PFJ anatomic abnormalities and combinations of intact knees in patients younger than 18 years suffering from acute traumatic first-time unilateral LPD.

The purpose of this study was to identify key differences and correlations of the PFJ morphology between intact and contralateral injured knees in a first-time traumatic LPD population aged under 18 years.

The null hypothesis of our work was that there are no differences and correlations of the PFJ morphology between intact and contralateral injured knees in a first-time traumatic LPD population aged under 18 years.

## 2. Materials and Methods

The study design was approved by the medical ethics committee of our institution, and written parental consent to participate in the investigation was obtained from all patients.

Consecutive patients younger than 18 years admitted to the emergency department of our hospital between 2012 and 2016 with evident first-time acute LPD requiring reduction were involved in the prospective investigation [16,17].

Exclusion criteria of the patients in this study included history of bracing or harnessing of the lower extremities in infancy, leg bone fractures or dislocations, prior surgery or patellar instability symptoms, and non-traumatic patellar dislocation of the intact or injured contralateral knee, as well as conditions associated with serious neuromuscular or congenital diseases.

At initial assessment after immediate arrival to the emergency department, all patients included in the study underwent the same clinical orthopedic examination and radiographic evaluation (anteroposterior and lateral views) of the injured knee, followed by dislocated patella reduction under anesthesia [2,3,6,14,15,16].

All patients included in the study underwent the same post-reduction radiographic evaluation of the intact and injured contralateral knees where the pain was properly suppressed. The X-ray tube was positioned at a one-meter distance from the knee, and standard standing anteroposterior, lateral with 30° knee flexion, and Merchant radiographs of both knees were obtained. The Merchant views were performed with the knee in 40° flexion and the X-ray tube inclined at 30° caudad [2,3,6,14,15,16].

All images were analyzed independently by two radiology consultants experienced in musculoskeletal radiography. Both reviewers were unacquainted with the results of previous image interpretations and were blinded to patient information and side of injury. In the event of disagreement, the images were then reviewed to reach a consensus. Both reviewers measured the following parameters.

Patellar height was measured on a plain lateral radiograph using the Blackburne–Peel method where the ratio of the articular surface length of the patella to the height of the lower pole of the articular surface above a tibial plateau line was calculated. Patella alta (PA) was assumed when the Blackburne–Peel index (BPI) was >1 [3].

The absence or presence and severity of the TD was classified on the lateral knee with skyline (Merchant) patellar radiographs using the Dejour classification. The femoral sulcus angle (FSA) was defined as the angle formed between lines joining the highest points of the bony medial and lateral condyles and the lowest bony point of the intercondylar sulcus on the Merchant view. An FSA measuring >145° was classified as TD [11,12], as shown in Figure 1.

Patella morphology was classified on axial images according to Wiberg [18].

The classic demographic and anthropometric data including patient height, weight, and body mass index (BMI) as well as its percentiles were calculated and recorded.

According to the international scientific literature, patients younger than 14 years at the time of primary LPD have the highest incidence of recurrent patellar dislocations, presenting a unique challenge to the orthopedic surgeon [1,18]. In order to identify key differences and correlations of the PFJ morphology between intact and contralateral injured knees in a first-time traumatic LPD population aged under 14 years, the patients were divided into two groups. There were 16 patients under 14 years of age and 42 older than 14 years.

In order to establish and compare the possible key differences, correlations between the FSA and patella with high anatomical variations of intact knees in the PFJ anatomy of patients with more (combination of the PA with SFS) and less (only PA) dysplastic injured knees in the two groups of subjects were examined. Group A had 26 patients with a combination of PA and SFS (more dysplastic PFJ anatomy) of the injured knee, while Group B group consisted of 24 patients with PA without TD of the injured knee.

### Statistical Analysis

PA and abnormal SFS were defined as the primary parameters for the study sample-size calculation. Based on earlier publications, the mean rates of PA and abnormal SFS were 60.12% and 63.3% in first-time LPD patients and 17.4% and 7.7% in control patients, respectively [5,7,9,10,11,14]. To detect a difference of the PA rate of 42.7% among patients with intact and contralateral injured knees under the assumption of a study power of 90% and a type I error (α) of 0.05, at least 25 children and adolescents (50 knees according to our research methodology) were needed in each group. In order to detect a difference of the abnormal SFS rate of 55.6% between patients with intact and contralateral injured knees, the sample size requirement was calculated with use of a study power of 95% and a type I error (α) of 0.05. This resulted in a required sample size of 16 patients (32 knees) per group, and this was increased by 40% to account for attrition. Accordingly, 23 patients (46 knees) were recruited, with 23 knees per group. However, we included 58 (116 knees) patients in our study.

The data are depicted as mean ± standard deviation (SD) for each variable. To test the normality of the variances, the Shapiro–Wilk W test was used. Student’s t-test was used to investigate differences between mean values. The Mann–Whitney U test was used to compare variables with an asymmetric distribution. Differences in the two-way tables were determined with the Pearson χ^2^ test with Yates’ continuity correction or the Fisher exact test when the expected cell count was less than five. The Pearson correlation coefficient (PCC) was used to examine the strength and direction of the monotonic relationship between two variables. The level of significance was 5% (*p* (two tailed) < 0.05) for all tests; values being significant at a 1% level (*p* (two tailed) < 0.01) were defined as highly significant.

Binomial logistic regression models were generated to ascertain the effects of patient sex, age, and anthropometrics (height, weight, BMI, contralateral injured knee BPI, uninjured knee BPI, and FSA) measurements on the likelihood that patients had SFS of the contralateral injured knees. The goodness of fit of our models was assessed with the Hosmer–Lemeshow test, for which *p* > 0.05 signified a good fit. A kappa statistic was used to assess inter-observer agreement for X-ray image studies to depict the PA by calculation of the Blackburne–Peel index and to define the FSA. The kappa value was interpreted as follows: poor agreement, <0.2; fair agreement, 0.2–0.4; moderate agreement, 0.4–0.6; good agreement, 0.6–0.8, and very good agreement, 0.8–1. Kappa analysis of X-ray image determinations between two blinded observers yielded values of 0.783 and 0.893 for undamaged patient knee PA and FSA, as well as 0.84 for the PA and 0.91 for the FSA of the injured patient knees, indicating good and very good concordance, respectively.

## 3. Results

The study population included 35 girls (60.3%) with a mean age of 15.1 years (SD = 1.51; range 11–17 years) and 23 boys (39.7%) with a mean age of 16.1 years (SD = 1.1; range 14–18 years) with 58 injured knee joints (35 (60.3%) left, 23 (39.7%) right). The mean height, weight, and BMI values of the patients were 1.71 m (SD = 0.1; range 1.5–1.94 m), 66.9 kg (SD = 12.3; range 46–100 kg), and 22.8 m/kg^2^ (SD = 3.8; range 15.3–37.2 m/kg^2^), respectively. In addition, the mean height, weight, and BMI percentiles of the patients were 73.8 (SD = 32.4; range 5.6–99.8), 73.9 (SD = 23.3; range 4.6–99.8), and 66 (SD = 25.2; range 0.1–99.4).

SFS was found in 24.1% (14 of 58) of patients’ uninjured knees and in 50% (29 of 58) of contralateral injured knees (*p* < 0.05).

The mean FSA was 139.8° (SD = 8.3; range 115.8° to 164.5°) of the uninjured knees and 143.5° (SD = 7.6; range 121° to 157.1°) of the contralateral injured knees (*p* < 0.05).

The frequency of PA was 84.5% (49/58) according to the criterion of Blackburne–Peel et al. of the uninjured knees and 86.2% (50/58) of the contralateral injured knees. The mean BPI was 1.2 (SD = 0.2; range 0.8 to 1.7) and 1.3 (SD = 0.3; range 0.8 to 2), respectively (*p* > 0.05).

There were statistically significant positive correlations (SSPC) between the FSA and BPI of the patients’ uninjured (*r* = 0.37; *p* < 0.005) and contralateral injured knees (*r* = 0.33; *p* < 0.05).

Binomial logistic regression analysis demonstrated that the most pertinent risk factors for SFS of the uninjured knees were contralateral injured knee FSA and uninjured knee BPI (*p* < 0.05). The significant risk factors for SFS of the contralateral injured knees were uninjured knee FSA and the patient’s height (*p* < 0.05).

The frequency of abnormal SFS of uninjured knees was 26.1% in boys and 22.8% in girls. We noted that 12 of 23 boys (52.2%) and 18 of 35 girls (51.4%) had abnormal SFS of the contralateral injured knees.

Overall, 5 of 16 children under the age of 14 (31.3%) and 9 of 42 children over the age of 14 years (21.4%) had abnormal SFS of the uninjured knees. In addition, 10 of 16 children under the age of 14 (62.5%) and 19 of 42 children over the age of 14 (45.2%) had abnormal SFS of the contralateral injured knees. We observed a statistically significant difference between the frequencies of abnormal SFS of the uninjured and contralateral injured knees in children older than 14 years of age and girls (*p* < 0.05). There were SSPCs between the FSA of the uninjured and contralateral injured knees in both gender and age groups of children (*p* < 0.05) (Table 1).

We observed PA of the uninjured knee in 21 of 23 boys (91.3%) and in 27 of 35 girls (77.1%) (*p* > 0.05). PA of the contralateral injured knees was found in 21 of 23 boys (91.3%) and in 29 of 35 girls (82.9%) (*p* > 0.05). PA of the uninjured knees was established in 15 of 16 children under the age of 14 (93.8%) and in 33 of 42 children over the age of 14 (78.6%) (*p* > 0.05). In addition, in 15 of 16 children under the age of 14 (93.8%) and in 35 of 42 children over the age of 14 (83.3%) PA was found in contralateral injured knees (*p* > 0.05).

There were SSPCs between the BPI of the uninjured and contralateral injured knees in both gender and age groups of children (*p* < 0.05) (Table 2).

There were SSPCs between the FSA and BPI of the >14-year-old patients’ uninjured (*r* = 0.32; *p* < 0.05) and contralateral injured knees (*r* = 0.44; *p* < 0.005).

The most common PFJ anatomic abnormality of injured knees was combination of SFS and PA. In contrast, high-riding patella was the most common abnormality of uninjured knees followed by the combination of PFJ dysplasia described above (Table 3).

The features of the FSA and the vertical position of the patella of the uninjured knees in patients with one (PA) and two (PA and SFS) abnormal PFJ morphological factors in their contralateral injured knees are provided in Table 4 and Table 5. There were no significant differences in patient characteristics (age, gender, height, height percentiles, weight, weight percentiles, BMI, BMI percentiles) between the two study groups.

SSPCs between the FSA and BPI of uninjured knees of the patients from the A group were found (*r* = 0.6; *p* < 0.001). There were likewise SSPCs between the FSA and BPI of contralateral injured knees of the Group B patients (*r* = 0.48; *p* < 0.05).

In the present study, the majority of patients were noted to have type B patella morphology (*n* = 50, 86.2%) according to Wiberg, while fewer patients had type C (*n* = 6, 10.4%) or type A (*n* = 2, 3.5%).

## 4. Discussion

The main findings of this study were that SSPCs existed between the FSA and PA of both gender and age groups of patients’ uninjured and contralateral injured knees, and SSPCs between the FSA and PA of uninjured knees were higher in patients with a more dysplastic (combination of PA with SFS) injured knee PFJ anatomy compared to patients with less aberrated (only PA) PFJ of the injured knees. It is very important to learn these relationships because the high-riding patella engages the femoral trochlea at greater degrees of knee flexion, and the SFS may allow the patella to migrate out of the trochlear groove as the quadriceps contracts and the knee extends from a flexed position that can lead to LPD [2,3].

There is no doubt that PFJ biomechanics are well established, but the etiology of aberrations is still the subject of much discussion. It is possible that there may be an integration of the relationship between the patella and the trochlear morphological development in the genome during the course of evolution, but some authors hypothesize that this represents a result of orthostatism and biped walking [13,14,18]. Under these circumstances children and adolescents experiencing an acute first-time LPD should have more frequent geometric morphological alterations in the intact knee PFJ compared with the total population data. Previous studies have shown PA occurs in 48–80% of patients who have dislocated their patella but in only 5.2–22.6% of asymptomatic healthy subjects [5,7,9,10,11,14]. The abnormal SFS rates range from 57.9% to 68.3% in first-time LPD patients and from 4.3% to 13.1% in control asymptomatic patients according to data from different studies [5,7,10,11].

When considering our study results, we found that abnormal SFS occurred in 24.1% of our patients’ uninjured knees according to the criterion of Dejour et al. (FSA of >145°) and in 50% of contralateral injured knees. In our study, PA was observed in 84.5% of patients’ uninjured and in 86.2% of contralateral injured knees. The combination of PA and SFS was found in 22.4% of uninjured and in 44.8% of the contralateral injured knees in our patient population. The data of our study slightly support the provision that children and adolescents who experienced an acute primary LPD have a more dysplastic anatomy of uninjured knee PFJ compared with the general population.

To the best of our knowledge, this is the first study focusing on the elucidation of uninjured knee PFJ morphology in patients under the age of 18 years and on the identification of key differences in their contralateral injured knees after traumatic first-time LPD. Thus, a straightforward comparison of our data with published research results is limited, but some trends can be identified.

Seeley et al., in a population of children and adolescents with acute first-time LPD, found that 33% of patients had normal PFJ anatomy [1]. The authors claimed that the remaining 67% demonstrated varying degrees (type A, 23%; type B, 28%; type C, 11%; and type D, 5%) of TD according to Dejour [1]. The distributions of the TD were similar across age groups and sex [1]. In the study, 62.2% of the patients had a normal subchondral sulcus angle <145 degrees; 46% of these patients had a subsequent increase in articular sulcus angle >145 degrees, qualifying them as dysplastic patients. PA was observed in 27% of the observed patients. Seeley et al. concluded that the majority of patients had type B morphology (73%), whereas fewer patients had type A (16%) or type C (10.8%) morphology of the patella according to Wiberg.

In 2013, Kohlitz et al. analyzed the three major risk factors (TD, PA and abnormally lateralized tibial tuberosity) of PFJ instability in a patient population with LPD and in an age- and sex-matched control group using MR imaging (MRI) [10]. The authors included 372 knees, 186 in LPD and 186 in the control group [10]. Sixty-one dislocators (36.1%) had concomitant TD and PA [10]. Forty-five patients (26.6%) had TD and no other risk factors [10]. Kohlitz et al. claimed that 25 individuals (14.8%) in the LPD group had none of the three risk factors and that in their control group 72.4% had no anatomical risk factors; 44 patients (25.9%) had at least one risk factor, and 3 patients (1.8%) had two risk factors. No subjects in the control group had all three risk factors [10].

Arendt and co-authors claimed that the main findings of their study were that 87% of patients with primary LPD had one or more anatomic risk factors, and PA with TD were the two most common factors as determined by MRI [11]. Arendt et al. detailed that within a single individual, PA had the highest representation when there was just one risk factor (47.9%), followed by trochlear aberration (39.6%). The combination of trochlear and patella high aberrations was established in 71% of patients who had two risk factors [11].

Palmu et al. studied a group of 71 children and adolescents with acute LPD [15]. The mean age of these patients was 13 years. SFS was found in 71% of patients, and PA was present in 66% of the affected knees [15]. PFJ instability of the contralateral knee was found in 48.4% [15]. Thirty-three patients (46%) had a positive family history of LPD [15].

In 2015, Zheng et al. investigated 127 knees of 73 (57.5%) girls and 54 (42.5%) boys with a mean age of 14.1 ± 4.5 years, and 96.9% of these patients after LPD had medial patellofemoral ligament disruption. Normal PFJ anatomy was observed in 54 (43.9%) of 123 patients with medial patellofemoral ligament tears [17]. In the remaining patients, the femoral TD was categorized as type A in 25.2% of cases, type B in 16.3%, type C in 10.6%, and type D in 4.1% of cases [17].

Seeley et al. included 46 patients with displaced osteochondral fractures [16] in their study. There were 32 male and 14 female patients, with a mean age of 14.6 years. PA was observed in 24% of the patients. The mean subchondral sulcus angle was 156.15 ± 8.75 degrees. Twenty-nine patients (63%) had subchondral sulcus angles >145 degrees, classifying them as dysplastic [16]. Seeley et al. reported that 72% of their patients with osteochondral injury had evidence of abnormal PFJ morphology. In this study group, the majority of patients were noted to have patella morphology according to Wiberg type B (76%), while fewer patients had type A (8.7%) or type C (15.2%) [16].

The strengths of the study are its prospective nature, sufficient sample size, strict inclusion of undoubtedly acute primary LPD patients, and strict exclusion criteria. A thorough description of demographics, anthropometric data, and geometric morphological alterations of the intact and contralateral injured knee PFJ in patients with first-time LPD was another strength of this research.

The study did have some limitations because patient examination in the emergency department was limited to the performance of their knees in plain radiographs. Imaging of the PFJ in LPD is essential to the accurate diagnosis of the pivotal causes and their adequate treatment. In acute cases, plain radiography is sometimes the only element used to provide diagnosis. A standard series of radiographs including standing anteroposterior, lateral with 30° knee flexion, and Merchant views, however, is still considered the first line imaging modality. It is important to obtain a true lateral radiograph with symmetric overlap of the medial and lateral femoral condyles. The lateral radiograph is the most useful in determining the presence and degree of PA. We chose for this purpose the Blackburne–Peel index, which relies on more consistent osseous landmarks. The Merchant view is useful to assess patellar tilt, subluxation, and FSA. The FSA in our study was measured from the highest points on the medial and lateral femoral condyles with the apex at the deepest point of the intercondylar sulcus. As outlined above in our study, we, like Neyret and co-authors, selected consistent osseous landmarks for the PFJ morphology measurement [14]. Neyret et al. reported and confirmed that PA can be equally well measured by MRI as compared to X-ray [14]. The authors found no significant difference between X-ray and MRI measurements of the patellar tendon length in a comparison of 42 knees with a history of patellar dislocation with 51 control knees [14]. Neyret et al. observed excellent correlations between the radiological and MRI indices for both groups of patients [14]. The simplicity of the X-ray evaluation methodology in our study reflects high interobserver reliability. There was an almost perfect interobserver reliability for both the PA and FSA (0.783 and 0.893) of the patients’ intact knees, as well as for the PA and the FSA (0.84 and 0.91) of the patients’ contralateral injured knees.

The data of this study could allow physicians to identify patients at higher risk of developing LPD of the contralateral knee and may prompt the consideration of preventive measures.

## 5. Conclusions

There were statistically significant positive correlations between the femoral sulcus angle and patella alta in both gender and age groups of patients with intact and contralateral injured knees. The statistically significant positive correlations between the femoral sulcus angle and patella alta of intact knees were higher in patients with more dysplastic patellofemoral joint anatomy (patella alta and trochlear dysplasia) of the injured knees compared to the patients with only patella alta of the injured knees.

## Figures and Tables

**Figure 1 medicina-57-00206-f001:**
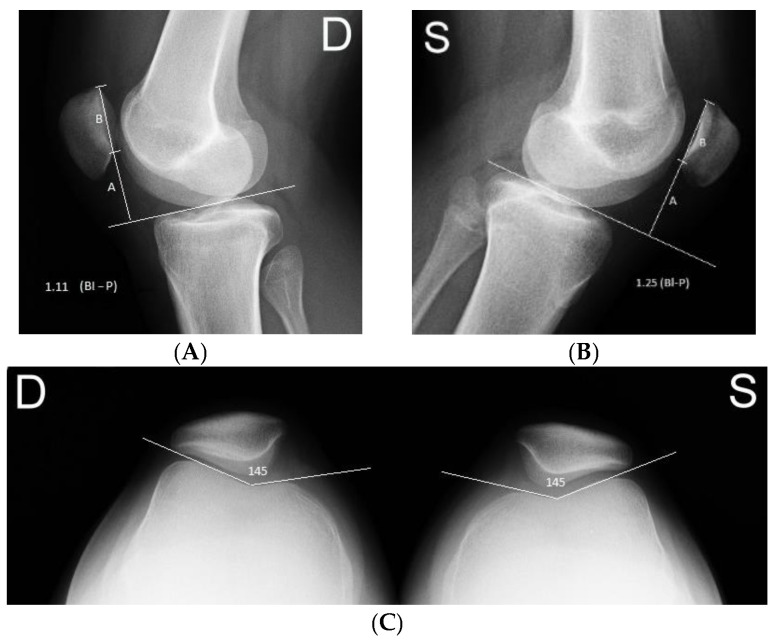
A 16-year-old girl with right side first acute lateral patellar dislocation (LPD). (**A**,**B**) Lateral radiograph of both knees demonstrating the Blackburne–Peel Index (BPI), with the ratio of the articular surface length of the patella (labeled as (**B**)) over the length of the perpendicular line drawn from the tangent to the tibial plateau to the inferior pole of the articular surface of the patella (labeled as (**A**)). A BPI more like 1 is considered patella alta. (**C**) Axial Merchant radiographic view demonstrating the femoral sulcus angle (FSA). (D—"dexter”—Latina for right side; S—"sinister”—Latina for left side). The FSA is the angle between the medial and lateral trochlear facets.

**Table 1 medicina-57-00206-t001:** Overview and statistical correlation of the uninjured and contralateral injured knee FSA in accordance with patient gender and age.

	Knee	Uninjured	Contralateral Injured
Demographic Factors			
Boys (*n* = 23)		141.4 ± 6.8	144.2 ± 6.1
		*r* = 0.61; *p* < 0.05	
Girls (*n* = 35)		138.7 ± 9.1	143 ± 8.5
		*r* = 0.85; *p* < 0.001	
≤14 years (*n* = 16)		142.5 ± 6	146.8 ± 6.7
		*r* = 0.65; *p* < 0.05	
>14 years (*n* = 42)		138.7 ± 8.9	142.2 ± 7.6
		*r* = 0.8; *p* < 0.001	

**Table 2 medicina-57-00206-t002:** Overview and statistical correlation of the intact and contralateral injured knee BPI in accordance with patient gender and age.

	Knee	Uninjured	Contralateral Injured
Demographic Factors			
Boys (*n* = 23)		1.28 ± 0.22	1.3 ± 0.25
		*r* = 0.52; *p* < 0.05	
Girls (*n* = 35)		1.22 ± 0.23	1.28 ± 0.27
		*r* = 0.78; *p* < 0.001	
≤14 years (*n* = 16)		1.22 ± 0.24	1.26 ± 0.27
		*r* = 0.64; *p* < 0.05	
>14 years (*n* = 42)		1.3 ± 0.18	1.34 ± 0.24
		*r* = 0.7; *p* < 0.001	

**Table 3 medicina-57-00206-t003:** Radiographic patellofemoral joint (PFJ) anatomic abnormalities of intact and contralateral injured knees.

	Knee	Uninjured	Contralateral Injured	*p*
Roentgenographic Findings				
PA and SFS		13 (22.4%)	26 (44.8%)	*p* < 0.05
PA		36 (62.1%)	24 (41.4%)	*p* < 0.05
SFS		1 (1.72%)	3 (5.2%)	*p* > 0.05
Normal anatomy		8 (13.8%)	5 (8.62%)	*p* > 0.05

**Table 4 medicina-57-00206-t004:** Overview of the uninjured and contralateral injured knee FSA in accordance with number of anatomic alterations in the surface geometry of the PFJ.

	Knee	Uninjured	Contralateral Injured	*p*
Group				
A group (*n* = 26)		(38.46%) 144.1 ± 4.77	(100%) 149.1 ± 3.69	*p* < 0.05
B group (*n* = 24)		(0%) 135.67 ± 5.5	(0%) 138.92 ± 4.75	*p* < 0.05
		*p* < 0.05	*p* < 0.05	

**Table 5 medicina-57-00206-t005:** Overview of the uninjured and contralateral injured knees BPI in accordance with number of anatomic alterations in the surface geometry of the PFJ.

	Knee	Uninjured	Contralateral Injured	*p*
Group				
A group (*n* = 26)		(88.5%) 1.31 ± 0.23	(100%) 1.38 ± 0.22	*p* > 0.05
B group (*n* = 24)		(91.7%) 1.24 ± 0.18	(100%) 1.31 ± 0.23	*p* > 0.05
		*p* > 0.05	*p* > 0.05	

## Data Availability

The data presented in this study are available on request from the corresponding author.

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
