# Peer review of "First-Time Acute Lateral Patellar Dislocation in Children and Adolescents: What about Unaffected Knee Patellofemoral Joint Anatomic Abnormalities?"

_medicina, 2021, doi:10.3390/medicina57030206_

Round 1

Reviewer 1 Report

The overall quality originality and quality of the manuscript is good and i think is suitable for publication in the present form. 

I have a couple suggestions:

1) add images of a clinical case (imaging) and a scheme that represents the calculation of patellar heigth and Femoral sulcus angle according the methods described

2) In the introduction section please report the extended form of FA, PSA and the other abbreviation (not only in the abstract)

Author Response

Thank you for your constructive comments and advice. I have corrected the manuscript according to your recommendations.

Sincerely Rasa Simonaityte

Reviewer 2 Report

I have included sticky notes and comments on the PDF.

Author Response

(The authors gave the same response as above.)
